# A New Circular Single-Stranded DNA Virus Related with Howler Monkey Associated Porprismacovirus 1 Detected in Children with Acute Gastroenteritis

**DOI:** 10.3390/v14071472

**Published:** 2022-07-04

**Authors:** Fabiola Villanova, Flávio Augusto de Padua Milagres, Rafael Brustulin, Emerson Luiz Lima Araújo, Ramendra Pati Pandey, V. Samuel Raj, Xutao Deng, Eric Delwart, Adriana Luchs, Antonio Charlys da Costa, Élcio Leal

**Affiliations:** 1Laboratório de Diversidade Viral, Instituto de Ciências Biológicas, Universidade Federal do Pará, Belem 66075-000, Pará, Brazil; fvillanova@gmail.com; 2Secretary of Health of Tocantins, Palmas 77453-000, Tocantins, Brazil; flaviomilagres@uft.edu.br (F.A.d.P.M.); eu3rafael@gmail.com (R.B.); 3Public Health Laboratory of Tocantins State (LACEN/TO), Palmas 77016-330, Tocantins, Brazil; 4General Coordination of Public Health Laboratories of the Strategic Articulation Department of the Health Surveillance Secretariat of the Ministry of Health (CGLAB/DAEVS/SVS-MS), Brasília 70719-040, Distrito Federal, Brazil; emerson.araujo@saude.gov.br; 5Centre for Drug Design Discovery and Development (C4D), SRM University, Delhi-NCR, Rajiv Gandhi Education City, Sonepat 131029, Haryana, India; ramendra.pandey@gmail.com (R.P.P.); deanacademic@srmuniversity.ac.in (V.S.R.); 6Vitalant Research Institute, 270 Masonic Avenue, San Francisco, CA 94118, USA; xdeng@vitalant.org (X.D.); edelwart@vitalant.org (E.D.); 7Department Laboratory Medicine, University of California San Francisco, San Francisco, CA 94143, USA; 8Enteric Disease Laboratory, Virology Center, Adolfo Lutz Institute, São Paulo 01246-000, São Paulo, Brazil; driluchs@gmail.com; 9Instituto de Medicina Tropical da Faculdade de Medicina da Universidade de São Paulo, São Paulo 05403-000, São Paulo, Brazil; charlysbr@yahoo.com.br

**Keywords:** *Smacoviridae*, porprismacovirus, virus-like particles, CRESS DNA, metagenomics

## Abstract

Putative replication-associated protein (REP) and capsid-like (CAP) proteins are encoded by circular single-stranded DNA viruses (CRESS DNA), which have been found in samples from most eukaryotic groups. However, the details of these viruses’ life cycles and their significance in diseases have yet to be established. We presented and analyzed two full-length CRESS DNA genomes acquired from two children diagnosed with acute gastroenteritis (GI) in the northeast state of Tocantins, Brazil, using next-generation sequencing and a virus-like filtration approach. Both sequences (named SmaCV3BR08 and SmaCV3BR291) are closely similar to a prior CRESS DNA sequence discovered in the feces of a new world monkey (*Alouatta caraya*) from the United States in 2009 and termed Howler monkey-associated porprismacovirus 1 (Genbank ID: NC 026317). According to our comparative study, these porprismacovirus genomes deviate by 10% at the nucleotide level. For comparative reasons, the divergence between our sequences (SmaCV3BR08 and SmaCV3BR291) and a porprismacovirus recently identified in a human fecal sample from Peru is 37%. These data suggest that there is a great diversity of porprismacoviruses in South America, perhaps more than two species. In addition, the finding of closely related sequences of porprismacoviruses in humans and native monkeys highlights the zoonotic potential of these viruses.

## 1. Introduction

The extensive use of next-generation sequencing on a variety of biological samples revealed an enormous variety of new viruses [1,2,3]. Particularly, viruses recently identified and coined as circular, Rep-encoding ssDNA (CRESS DNA) has a huge diversity and were detected in samples of eukaryotic hosts [4,5,6,7,8,9,10,11,12,13,14]. Many of the species belong to the phylum *Cressdnaviricota* and were classified into two genera, eight orders, eleven families, and sixty-six genera; many of the species are uncultured and remain unclassified [15]. The genome is small circular single-stranded DNA and likely encodes two proteins: rolling-circle replication initiation protein (Rep) and the unique capsid protein (CAP). The host range of these viruses includes plants (*Nanoviridae* and *Geminiviridae*), fungi (*Genomoviridae*), algae (*Bacilladnaviridae*), and animals (*Circoviridae*). Members of the families *Smacoviridae* and *Redondoviridae* have been identified solely by metagenomics, and they likely infect animals [16].

REP-Based phylogenies indicate two large classes representing the phylum Cressdnaviricota: (i) the first class (*Repensiviricetes*) has a single order (*Geplafuvirales*) that includes members of the families *Geminiviridae* and *Genomoviridae*, and the unclassified clade CRESSV6 viruses; (ii) the second class (*Arfiviricetes*) has seven orders (*Baphyvirales*, *Cirlivirales*, *Cremevirales*, *Mulpavirales*, *Recrevirales*, *Rivendellvirales,* and *Rohanvirales*). The order *Cremevirales* comprises twelve genera and one family (*Smacoviridae*), including the Porprismacoviruses. There are at least 84 species in this family, which have been found in a wide range of animals [15,17].

Smacovirus (single-stranded DNA virus) is a type of single-stranded DNA virus. The genome is circular single-stranded DNA that is 2.3–2.8 kilobases long and contains two potential proteins (i.e., Rep and CAP).

Because viruses in this taxon were discovered in feces samples from insects and vertebrates using metagenomic methods, their host range and biology in cells remain unknown [16,17,18,19].

In the current study, we present the results of characterization and phylogenetic comparison of two sequences of the genera Porprismacovirus that were identified in fecal samples from children with acute gastroenteritis. These sequences are phylogenetically related to one sequence previously identified in a fecal sample from a howler monkey. Because all of these genomes were discovered using next-generation sequencing of feces, the natural host of these viruses cannot be determined. The fact that porprismacoviruses have been found in a variety of species shows that they can infect a wide spectrum of eukaryotic species. Our findings highlight the considerable genetic diversity of porprismacoviruses throughout South America, adding to the virus’s zoonotic potential.

## 2. Materials and Methods

### 2.1. Ethics Information

The survey was conducted in accordance with the 1975 Declaration of Helsinki (https://www.wma.net/what-we-do/medical-ethics-of-helsinki/ accessed on 10 March 2021), revised in 2013. The project was approved by the Committee of Ethics of the institutions involved (Faculty of Medicine, University of São Paulo (CAAE: 53153916.7.0000.0065), and Lutheran University Center of Palmas—ULBRA (CAAE: 53153916.7.3007.5516). There was no risk or harm to the children or their guardians; therefore, it was not necessary to apply the Informed Consent Term (ICF) in accordance with resolution 196/96 on research involving human beings—National Health Council (CNS)/Ministry of Health (MS), Brasília, 1996.

### 2.2. Study Population and Sample Collection

The current cross-sectional surveillance study was carried out in partnership with the Central Public Health Laboratory (LACEN) between 2010 and 2016 in the states of Tocantins (TO), Maranhão (MA), and Pará (PA), located in the north (TO/PA) and northeast (MA) of Brazil. Fecal samples were collected in 38 different municipalities. In this study, we analyzed 250 samples: 3 from Pará, 3 from Maranhão, and 244 from the Tocantins. A total of 232 stool samples were collected from children under the age of five, three children between the ages of six and eleven, three adolescents between the ages of twelve and seventeen, one young person between the ages of eighteen and twenty-three, four adults between the ages of 24 and 59, and two elderly people over the age of sixty. Because of a lack of information, inadequate data, or the use of abbreviations to identify patient names, clinical data regarding age were not available for 5 patients. It is noteworthy that all patients on admission had GI symptoms (i.e., diarrhea, nausea, vomiting, and fever).

### 2.3. Sample Screening

The samples were initially delivered to LACEN/TO with an epidemiological investigation record that included the subjects’ demographic data (age, sex, and collection date) as well as clinical information (signs and symptoms). Subsequently, the samples were stored at −20 °C and forwarded to the USP Institute of Tropical Medicine of the University of São Paulo (IMT-USP). The aim of the original study was to search human feces for the presence of common and potential novel viral enteric pathogens in rural and low-income urban areas in northern Brazil using NGS techniques. However, the NGS surveillance applied also offered the opportunity to study commensal viruses, bacteriophages, and plant viruses, among others.

### 2.4. Viral-Like Particle Metagenomics

The protocol used to perform the deep sequencing was a combination of several protocols applied to viral metagenomics and/or virus discovery, according to the procedures described previously [20,21,22]. Briefly, 50 mg of each human fecal sample was diluted in 500 μL of Hank’s Buffered Saline (HBSS) and added to a 2 mL impact-resistant tube containing lysis matrix C (MP Biomedicals, Santa Ana, CA, USA) and homogenized in a FastPrep-24 5G Homogenizer (MP Biomedicals, USA). The homogenized sample was centrifuged at 12,000× *g* for 10 min, and approximately 300 μL of the supernatant was percolated on a 0.45 μm filter (Merck Millipore, Billerica, MA, USA) to remove bacterial and eukaryotic cells. Approximately 100 μL of PEG-it virus precipitation solution (System Biosciences, Palo Alto, CA, USA) was added to the filtrate, and the tube contents were gently homogenized, followed by incubation at 4 °C for 24 h. After the incubation period, the mixture was centrifuged at 10,000× *g* for 30 min at 4 °C, and the supernatant (~350 μL) was discarded. The granulate, rich in virus-like particles (VLPs), was treated with a combination of nuclease enzymes (TURBO DNase and RNase Cocktail Enzyme Mix—Thermo Fischer Scientific, Waltham, MA, USA; Baseline ZERO DNase DNase—Epicenter, Madison, WI, USA; Benzonase—Darmstadt, Darmstadt, Germany, and RQ1 DNase-Free DNase and RNase A Solution—Promega, Madison, WI, USA) to digest unprotected nucleic acids. The resulting mixture was subsequently incubated at 37 °C for 2 h. Viral nucleic acids were then extracted using the ZR and ZR-96 viral DNA/RNA kits (Zymo Research, Irvine, CA, USA) according to the manufacturer’s instructions. cDNA synthesis was performed with an AMV reverse transcription reagent9 (Promega, Madison, WI, USA). Synthesis of second-strand cDNA was performed using a large fragment of DNA polymerase I (Klenow) (Promega). Subsequently, a Nextera XT Sample Preparation Kit (Illumina, San Diego, CA, USA) was used to build a DNA library, which was identified by a double barcode. The library was then purified using the ProNex^®^ size-selective purification system (Promega, WI, USA). After ProNex^®^ purification, the amount of each sample was normalized to ensure equal representation of the library with the pooled samples using the ProNex^®^ NGS Library Quant Kit (Promega, WI, USA). For the size range, Pippin Prep (Sage Science, Inc., Beverly, MA, USA) was used to select a 300 bp tablet (range 200 to 400 bp), which excluded very short and long fragments from the library. Prior to cluster generation, libraries were quantified again by qPCR using the ProNex^®^ NGS Library Quant Kit (Promega, WI, USA). The library was sequenced in depth using a Hi-Seq 2500 Sequencer (Illumina, CA, USA) with 126 bp ends.

### 2.5. Bioinformatics Analysis

Deng et al. [23] previously described a protocol for bioinformatics analysis. Bowtie2 was used to eliminate non-viral sequences (i.e., human, bacterial, and fungal sequences). Only one random copy of duplicate reads was kept in reading bases with a Phred quality score of less than 20. The Ensemble Assembler pipeline was used to assemble the remaining reads from scratch for each individual sample. BLASTx was used to link the contigs generated for eukaryotic and prokaryotic viral sequences to the RefSeq virus protein database. BLASTx was used to classify sequence hits with a lower E value (E value 0.001) to viruses than to non-viral proteins into their corresponding viral family and genus. Following virus identification, viral contigs were used to query the reference sequences, which were then used to map the entire genome using Geneious R9 software (Biomatters Ltd. L2, 18 Shortland Street, Auckland, New Zealand).

### 2.6. Alignments and Annotation

The resulting contigs were subjected to a modified protein blast search using Ugene software [24] to identify novel members of the CRESS-DNA viruses. Based on the best results (best hits) from the BLASTx search, genomes of pecovirus and other related viruses were chosen for further analysis. Next, full or nearly full genomes were aligned using MAFFT software [25]. Genome annotation was performed using Gatu software [26], and the sequence NC_026317 was used as a reference.

### 2.7. Genetic Variability

The maximum composite likelihood model with gamma correction and bootstrap with 100 repetitions was used to calculate genetic distance and its standard error. The MEGA software (Version X) was used to calculate distances [27]. We utilized a pairwise method implemented in the program SDT [28] to estimate sequence similarity. Algorithms implemented in MUSCLE were used to estimate the similarity alignments of each unique pair of sequences [29]. The program then utilizes the NEIGHBOR component of PHYLIP to construct a tree [30] after computing the identity score for each pair of sequences. All sequences were arranged in the rooted neighbor-joining phylogenetic tree according to their apparent evolutionary relatedness. In a graphical interface, the results are displayed as a frequency distribution of pairwise identities.

### 2.8. Phylogenetic Analysis

Phylogenetic trees of CRESS-DNA virus genomes were constructed using the maximum likelihood approach. In order to obtain reproducible results and provide greater reliability of the clustering pattern of trees, the statistical support of branches was evaluated by the approximate likelihood ratio test (aLRT). Trees were inferred using the FastTree [31] software and the GTR model, plus gamma distribution and the proportion of invariable sites were used. Selection of the best model was made according to the likelihood ratio test (LRT) implemented in the Modeltest software [32].

## 3. Results

### 3.1. Blast Search and Sequence Characterization

We found reads of nanovirus (family: *nanoviridae*, genera nanovirus) in one individual and reads of circovirus (family: *circoviridae*) in five individuals. Reads of smacovirus were detected in three individuals (family: *smacoviridae*, genera porprismacovirus), and two full genomes were assembled. These genomes were deposited in the genbank (Ids: ON623842 and ON616517). Details of these individuals and also the description of all viruses detected in their samples were summarized in a previous publication [33,34]. After the de novo assembled of contigs, we used blastn to characterize the viral sequences. We found two CRESS-DNA virus contigs related to porprismacovirus, named SmaCV3BR08 (ON623842) and SmaCV3BR291 (ON616517). The best hits of the blastN search (E value < 1 × 10−68) indicated that both sequences were highly similar to a CRESS DNA sequence detected in the fecal sample of a new world monkey (*Alouatta caraya*) from the USA in 2009 and designated as Howler monkey associated porprismacovirus 1 (Genbank ID: NC_026317). The similarity of SmaCV3BR08 and SmaCV3BR291 with these sequences detected in a monkey was 85.54% and 87.98%, respectively. It is important to mention that the second-best hit was a CRESS-DNA sequence identified in a pig in Vietnam (Genbank ID: MH111140). The similarity was 74.84% and 74.92%, respectively, with SmaCV3BR08 and SmaCV3BR291. (This gap in the similarity between the first and second hits indicates the lack of homolog sequences in the databank, it also suggests that there is a high divergence in CRESS-DNA sequence detected in human samples (see Table 1). Next, we used the orffinder tool (https://www.ncbi.nlm.nih.gov/orffinder/ accessed on 5 May 2022) to identify open reading frames (ORFs) in our sequences; more than 28 ORFs were detected in each sequence. There are two cognate proteins in the Blast reference proteins database corresponding to these ORFs: one corresponding to the CAP protein and the other corresponding to the REP protein of CRESS-DNA viruses. In both cases, the most similar proteins were those of the Howler monkey-associated porprismacovirus 1.

### 3.2. Genome Annotation

The sequence of the Howler monkey-associated porprismacovirus 1 (NC_026317) was used as a reference to perform a detailed annotation of the Brazilian sequences (see the annotation of this sequence in Appendix A). The SmaCV3BR08 and SmaCV3BR291 contained 2799 and 2937 bases with a CG content of 49.79% and 49.49%, respectively. We found two potential ORFs in the opposite orientation and the size and location of putative proteins CAP and REP with 1119bp and 501bp, respectively (Figure 1). Interestingly, the putative REP protein of the Howler monkey-associated porprismacovirus 1 (NC_026317) is composed of two CDS, one with 162bp and the other with 534bp, separated by an intron of 225bp. We found only one CDS in the REP protein, and no splicing signals were detected in the genomes of SmaCV3BR08 and SmaCV3BR291. The Brazilian sequences also have the canonical nonamer TAGTGTTAC, which forms a stem-loop that indicates the replication origin in CRESS-DNA viruses (indicated in a magenta rectangle in Figure 1). The loop is located at positions 1809 to 1817 and 1898 to 1906, respectively, in the genome of SmaCV3BR08 and SmaCV3BR291. We also found the inverted repeats AAGCACAGTTCAGGCAAT and ATTGCCTGAACTGTGCTT.

These inverted repeats are 44 nucleotides apart and may play a role in rolling circle genome replication. Their locations in the genomes are shown in Figure 1. (green rectangles). The proteasome non ATPase 26S subunit (EAKEGQYWKSSDRVDNLIQRFGEFRPNQKRAIQALRATNDREVLVWYDEGGNVGKSWF), situated between residues 80 and 137, and the DEXH-box helicase domain of RecQ family proteins (DNLIQRFGEFRPNQKRAIQAL), located between residues 94 and 14, were also discovered in REP protein.

### 3.3. Nucleotide Variability in Members of CRESS DNA Viruses

In order to better evaluate the identity of our sequences, we performed a similarity analysis, comparing the complete genomes of Brazilian sequences with other members of smacovirus. We selected smacovirus sequences based on their phylogenetic relatedness (see phylogenetic analysis below) and found that the nucleotide similarity of SmaCV3BR08 and SmaCV3BR291 with the smacovirus of Howler money was 87–88%, respectively. It is interesting that viruses detected in Brazil have less than 65% similarity compared to other smacoviruses in Table 1. For illustrative purposes, the similarity of SmaCV3BR08 and SmaCV3BR291 with another porprismacovirus detected in a human fecal sample from Peru is less than 60%. This may indicate the presence of very divergent porprismacoviruses in Latin American countries.

### 3.4. Differences in Residues in the Proteins CAP and REP of Porprismacovirus

We also performed a comparison of the amino acid composition in the CAP and REP proteins between our sequences and the Howler monkey-associated porprismacovirus 1 (NC_026317). In the Cap protein, we found 12 differences in the amino acid composition between SmaCV3BR08 and SmaCV3BR291, 29 differences between SmaCV3BR291 and NC_026317, and 40 between SmaCV3BR08 and NC_026317 (upper panels in Figure 2). Likewise, in the REP proteins, we found 10 differences between our sequences and NC_026317 (lower panels in Figure 2).

### 3.5. Phylogenetic Tree of Complete Sequences of CRESS DNA Viruses

We performed a phylogenetic analysis comparing the complete genomes of Brazilian sequences with some members of smacovirus. The tree shows clusters composed of viruses that were detected in distinct hosts. For example, there is a cluster composed of viruses found in pigs, monkeys, and flies, as indicated by a gray arrow in Figure 2. Moreover, clusters include sequences from different geographical regions. For example, some porprismacovirus detected in humans from Peru and Botswana are in the same phylloclade (indicated by blue dots in Figure 3).

The topological structure of the tree indicates a lack of geographical or host structure. This might be related to the fact that smacoviruses are identified in fecal samples; thus, some of these viruses might infect the intestinal microorganisms or even the remaining dietary intake. The ancestral pattern of SmaCV3BR08 and SmaCV3BR291 indicated that the closest sequences are the smacovirus of howler money_NC026317 (Clade A in Figure 3). These sequences are very close, and their divergence is 10% (gray rectangle in Figure 3). Sequences of the closest clade (Clade B in Figure 3) diverge by 22% (gray rectangle in Figure 3). The genetic divergence between the sequences of clade A and clade B is 37%. Howler monkeys are endemic in Brazil. Although it may appear that this porprismacovirus infects both humans and monkeys, the fecal virome is primarily made up of viruses from the intestinal microflora.

## 4. Discussion

We reported the first identification and characterization of porprismacovirus in Brazil. The identified genomes had 2799 and 2937 bases with a similar genomic structure to the smacoviruses genomes, with ORFs for the REP and CAP proteins, the canonical smacovirus nonamer, and rolling circular replication motifs and helicase motifs. There was a high similarity (88%) with the sequence of the porprismacovirus NC_026317 previously identified in the USA in a fecal sample of a howler monkey. In addition to the nucleotide similarity, our sequences and the howler monkey porprismacovirus are also phylogenetically related. The circulation of different species has also been suggested by other authors who recently identified CRESS DNA in Cameroon in a sample of individuals with symptoms of gastroenteritis [35], curiously the same type of clinical sample used in the present investigation. More recently, CRESS DNA viruses of distinct families were identified in non-human primates from Zambia [35]. The authors note that some of these viruses increase their diversity by genome recombination. We found no signal of recombination in trees inferred with porprismacoviruses. Although we found topological incongruence between the trees inferred with CAP and REP proteins (Appendix A), no recombination was detectable in the complete genome of porprismacoviruses. Additionally, the clade composed of our sequences and the howler monkey porprismacovirus was unique in both trees.

When compared with CAP protein, REP protein is the most conserved at the amino acid level among CRESS DNA viruses. However, both proteins are divergent even among viruses identified in the same species. The overall divergence of REP and CAP in porprismacoviruses is 73% and 95%, respectively. In the comparison between our sequences and the Howler monkey-associated porprismacovirus 1 (NC_026317), we found that most differences in the amino acid composition were in the CAP protein. These amino acid changes were evenly distributed in the REP protein. In the CAP, protein changes were also distributed along with the protein, with some regions having clusters of substitutions (i.e., residues 113 to 116 and 248 to 259). Although most substitutions were conservative, some of them changed the hydrophobicity of the protein, such as K63I and T94V in REP and I113S and Q1234V in CAP. We also observed one substitution in Rep that modified the charge of the residue (i.e., E54K). Despite the high evolution indicated by insertions and deletions and sequence variability, there are conserved regions and motifs that indicate the active function of proteins of these viruses [16]. The complexity of CRESS DNA viruses, revealed through high throughput sequencing, has demonstrated their ability to spread in different environments and hosts [1,2]. Viral metagenomics has shown in recent years the description of eukaryotic ssDNA in healthy individuals, which may be part of the virome present in the gut of humans [4,9,18]. These viruses could influence the gut environment and the equilibrium of bacterial populations in the gut, which could also explain the occurrence of sequences of the pathogenic viral families in the intestinal lumen.

In addition, these viruses identified in our study are closely related to a virus that infects howler monkeys (*Alouatta caraya*). This kind of monkey is endemic in South America. Brazil has experienced unprecedented deforestation in the last four years, which is contributing to an extensive loss of habitat for many wild species. Consequently, encounters between humans and wild animals are becoming more common, and this can increase the chances of transmission of pathogens.

Metagenomic studies have important limitations, such as the absence of control samples. Another key limitation is the fact that fecal samples contain cells of the microflora and undigested material from the dietary intake. Although we used the VLP approach to minimize this kind of contamination, the link between the host virus is still uncertain. Furthermore, the samples were not examined using specific primers for polymerase chain reaction (PCR), which could increase the percentage of positive individuals. Furthermore, we have not investigated a large number of samples from people with varying clinical statuses to see if porprismacovirus can induce gut disorders, intensify the sickness caused by other pathogenic intestinal viruses, or if it is a normal part of the gut microbiota.

## 5. Conclusions

This study found the porprismacovirus, a CRESS DNA virus, in the feces of two infants in Brazil for the first time, implying that distinct species of this virus disseminate throughout South America. The finding of closely related sequences of porprismacoviruses in humans and native endemic monkeys from South America highlights the zoonotic potential of these viruses.

## Figures and Tables

**Figure 1 viruses-14-01472-f001:**
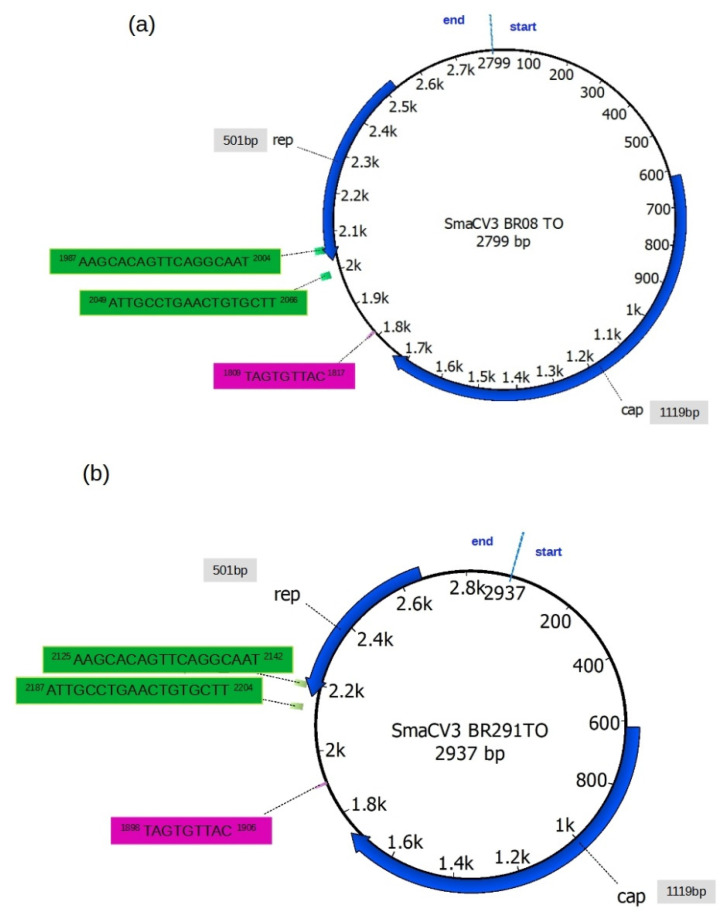
Genomic annotation of Brazilian porprismacoviruses. (**a**) Genomic map of the sequence SmaCV3BR08 and (**b**) map of the sequence SmaCV3BR291. Both sequences were annotated according to the predicted location of open reading frames of the reference sequence NC_026317 of CRESS-DNA viruses (Family: Cremevirales; Genera: Smacoviridae; species: Porprismacovirus). Rep denotes the replicase protein, and CAP denotes the capsid protein of these viruses. The sizes of Rep and Cap proteins are indicated in base pairs in the gray rectangles. The TAGTGTTAC loop is indicated in a magenta rectangle. Inverted repeats are indicated in green rectangles. The blue line perpendicular to the circular genome indicates the first nucleotide of the sequence.

**Figure 2 viruses-14-01472-f002:**
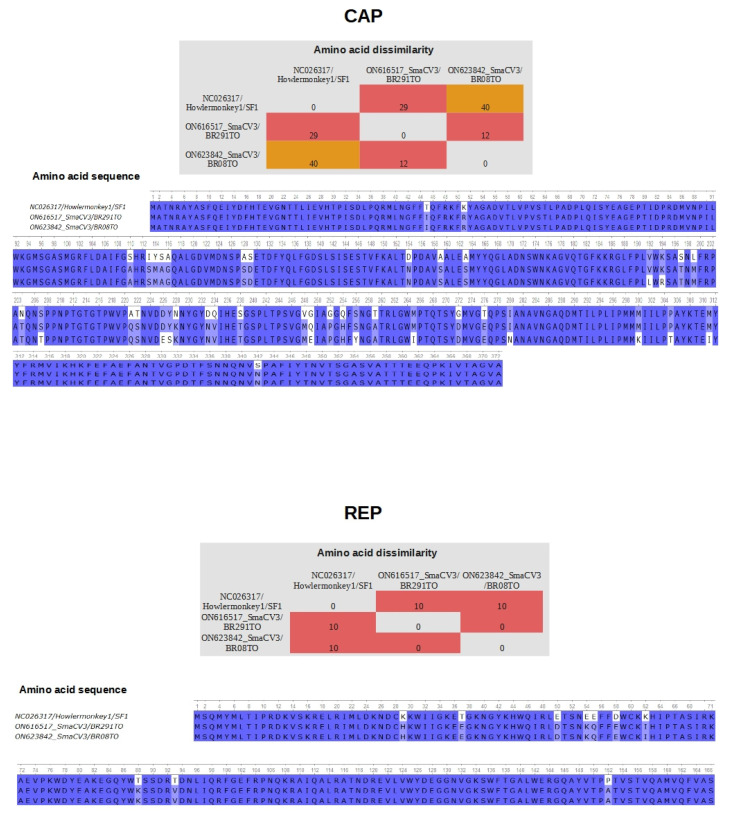
Amino acid composition of CAP and REP. Comparison of the predicted CAP and REP proteins between sequences generated in this study (SmaCV3BR08 and SmaCV3BR291) and the Howler monkey-associated porprismacovirus 1 (NC_026317). Upper panels show a matrix indicating number of distinct acid acids and the sequence of residues in the homolog region of the predicted CAP protein. The lower panels show the differences in amino acid composition and the sequences of predicted REP protein in Howler monkey-associated porprismacovirus, SmaCV3BR08 and SmaCV3BR291.

**Figure 3 viruses-14-01472-f003:**
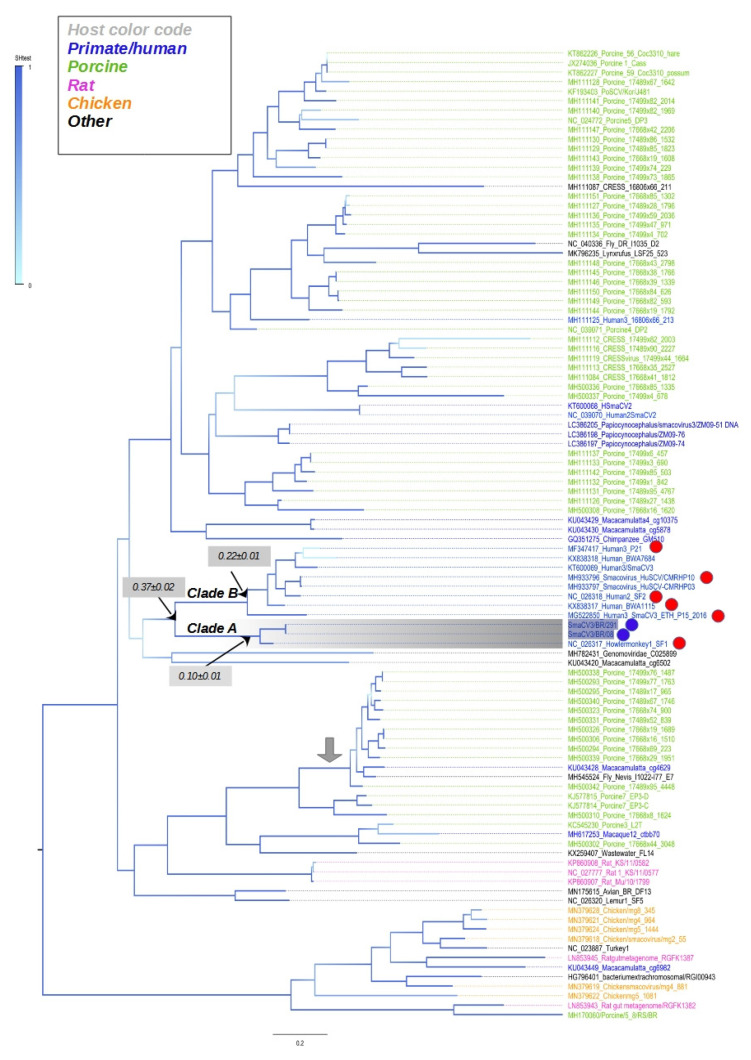
Phylogenetic analysis of complete genomes of smacoviruses. Maximum likelihood tree inferred assuming the GTR model. The tree is mid-rooted, and the statistical branch support (based on the Shimodaira–Hasegawa test) is indicated in a blue color scale. Lower horizontal bar indicates the nucleotide substitutions per site. Porprismacoviruses identified in this study are indicated by blue dots. Red dots indicate sequences used for nucleotide identity (Table 1 in the main text). The genetic divergence of some clades is indicated in the gray rectangles. Sequences are colored according to their hosts (upper left panel).

**Table 1 viruses-14-01472-t001:** Nucleotide identity of complete genomes of members of smacovirus.

Genbank ID (Genome Length in bp ^1^)	Host */Date/Location	SmaCV3BR08	SmaCV3BR291
		Identity	
NC026317 (2485)	*Alouatta caraya*/2009/USA	0.874	0.884
MG522850 (2511)	*H. sapiens*/2016/Ethiopia	0.650	0.643
KX838317 (2581)	*H.sapiens*/2012/Botswana	0.640	0.643
NC026318 (2529)	*P. troglodytes*/2012/USA	0.660	0.654
MH933796 (2538)	*H.sapiens*/2014/Cameroon	0.650	0.651
MF347417 (2560)	*H.sapiens*/?/Peru	0.644	0.646

* this refers to the species in which fecal samples were collected. ^1^ Base pairs.

## Data Availability

Not applicable.

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
