# Peer review of "A New Circular Single-Stranded DNA Virus Related with Howler Monkey Associated Porprismacovirus 1 Detected in Children with Acute Gastroenteritis"

_viruses, 2022, doi:10.3390/v14071472_

Round 1

Reviewer 1 Report

Advances in next generation sequencing methods have highlighted the presence and diversity of small, circular, single-stranded DNA (CRESS DNA) viruses in a variety of eukaryotic host samples. However, the details regarding the host range of these viruses, as well as their involvement in disease, remain poorly understood. In this study, Villanova et al. have isolated two full length CRESS-DNA genomes from two children diagnosed with acute gastroenteritis in Tocantins, Brazil and have employed a bioinformatics approach for further characterization. Comparative sequence analysis reveal that the two virus isolates (SmaCV3BR08, SmaCV3BR291) were most closely associated with Howler monkey associated porprismacovirus 1, and diverged by 37% when compared to another porprismacovirus recently isolated from human stool in Peru. The authors argue that these results highlight the genetic diversity of porprismacovirus in the South American region, as well as the broader zoonotic potential of these viruses. 

Major Comments: 

1) This work would benefit from a third figure that highlights the hotspots of sequence diversity between, at minimum, the two novel isolated genomes and the Howler monkey reference genome. This paper has gone into detail about the percent divergence of these two sequences against the most closely related sequences identified on Genbank, and additionally goes into detail regarding the overall sequence divergence of CAP and REP porprismacoviruses in the discussion (lines 316-319). The addition of a figure that directly compares these aspects would help to tie this information together, alongside new accompanying text.

2) Please improve the syntax and grammar of this manuscript. While the text can be read and understood, there are numerous issues with subject/verb agreement, singular/plural nouns, and extraneous articles throughout. While an extensive sweep of the text is warranted, examples include:

Line 77, "one sequences"

Line 199, "[one] individuals and reads of circovirus (family: circoviridae) in five individual"

Line 338, "this study found the porprismacovirus, a CRESS DNA virus, in the faeces of two"

Please address these in the changes.

Minor Comments:

1) Regarding Figure 1: Please standardize the label sizes, terminology and aesthetics between 1a and 1b. Additionally, it is difficult to read the white text in the green and magenta rectangles.

2) Regarding Figure 2: Please ensure that the highest possible resolution of this tree has been incorporated into the manuscript, as it is difficult to read the text on the branches.

3) Regarding Table 1: Please make sure that all of this table appears on the same page. I would also recommend implementing boarders around this table so that it's easier to parse information at a glance.

4) There is an inconsistent use of "fecal" vs. "faecal" throughout the text. Please standardize.

5) Line 205: Typo of "viruse"

6) Lines 211-214: It is difficult to parse the meaning of why it is important to note that the second best bioinformatic hit had respective sequence similarity of 74.84% and 74.92%. Is this to highlight that the the first hit is much higher at 85.54% and 87.89%? If so, please make this intent more clear.

7) Lines 214-216. This sentence is either missing some punctuation or should be made into two separate lines (e.g, new sentence at "More than 28 ORFs...")

8) Line 282: Reword "divergent divergence".

9) Line 287: Change "viroma" to "virome".

Author Response

Advances in next generation sequencing methods have highlighted the presence and diversity of small, circular, single-stranded DNA (CRESS DNA) viruses in a variety of eukaryotic host samples. However, the details regarding the host range of these viruses, as well as their involvement in disease, remain poorly understood. In this study, Villanova et al. have isolated two full length CRESS-DNA genomes from two children diagnosed with acute gastroenteritis in Tocantins, Brazil and have employed a bioinformatics approach for further characterization. Comparative sequence analysis reveal that the two virus isolates (SmaCV3BR08, SmaCV3BR291) were most closely associated with Howler monkey associated porprismacovirus 1, and diverged by 37% when compared to another porprismacovirus recently isolated from human stool in Peru. The authors argue that these results highlight the genetic diversity of porprismacovirus in the South American region, as well as the broader zoonotic potential of these viruses. 

Major Comments: 

1) This work would benefit from a third figure that highlights the hotspots of sequence diversity between, at minimum, the two novel isolated genomes and the Howler monkey reference genome. This paper has gone into detail about the percent divergence of these two sequences against the most closely related sequences identified on Genbank, and additionally goes into detail regarding the overall sequence divergence of CAP and REP porprismacoviruses in the discussion (lines 316-319). The addition of a figure that directly compares these aspects would help to tie this information together, alongside new accompanying text.

Resp: We created a diagram depicting the amino acid alterations in the CAP and REP proteins. We made the required changes to the manuscript as well.

2) Please improve the syntax and grammar of this manuscript. While the text can be read and understood, there are numerous issues with subject/verb agreement, singular/plural nouns, and extraneous articles throughout. While an extensive sweep of the text is warranted, examples include:

Line 77, "one sequences"

Line 199, "[one] individuals and reads of circovirus (family: circoviridae) in five individual"

Line 338, "this study found the porprismacovirus, a CRESS DNA virus, in the faeces of two"

Please address these in the changes.

Resp: We made the changes to the manuscript in order to increase the text's quality.

Minor Comments:

1) Regarding Figure 1: Please standardize the label sizes, terminology and aesthetics between 1a and 1b. Additionally, it is difficult to read the white text in the green and magenta rectangles.

Resp: We made changes to the manuscript in order to increase the quality of the text.

2) Regarding Figure 2: Please ensure that the highest possible resolution of this tree has been incorporated into the manuscript, as it is difficult to read the text on the branches.

Resp: We also enhanced the figures' quality and modified the font size.

3) Regarding Table 1: Please make sure that all of this table appears on the same page. I would also recommend implementing boarders around this table so that it's easier to parse information at a glance.

Resp: This table was changed accordingly.

4) There is an inconsistent use of "fecal" vs. "faecal" throughout the text. Please standardize.

Resp: This was updated throughout the text.

5) Line 205: Typo of "viruse"

Resp: This typo has been corrected.

6) Lines 211-214: It is difficult to parse the meaning of why it is important to note that the second best bioinformatic hit had respective sequence similarity of 74.84% and 74.92%. Is this to highlight that the the first hit is much higher at 85.54% and 87.89%? If so, please make this intent more clear.

Resp: Although there is no clear threshold in the percentage of similarity to classify CRESS-DNA virus, it is assumed that 90% of identity is enough. For example, the similarity between the sequences identified in our study is on average 99%, while they have 88% of identity with Howler monkey associated porprismacovirus 1 and 65% with the identified in Human CRESS-DNA virus fromPeru.

7) Lines 214-216. This sentence is either missing some punctuation or should be made into two separate lines (e.g, new sentence at "More than 28 ORFs…")

Resp: We have changed this sentence.

8) Line 282: Reword "divergent divergence".

Resp: We have changed this in the new version of the manuscript

9) Line 287: Change "viroma" to "virome".

Resp: This was changed in the manuscript

Reviewer 2 Report

Review:

A new circular single—stranded DNA virus related with 2 Howler monkey associated porprismacovirus 1 detected in 3 children with acute gastroenteritis

The authors identify and describe the genetic make up of two CRESS strains detected in in children with gastrentiritis. These strains are related to Howler monkey associated porprismacovirus. The manuscript is well written, and the data supports the hypothesis. My main comment is on the consistency between the different sections of the paper.

The authors should pay special attention to two things:

1.     In the discussion (lines 310 – 313) the authors comment on two trees based on the proteins of CAP and REP. However, in the results section the only tree discussed (Fig 2) is based on the whole genome. I didn’t see any supplementary data and if these trees were done it should be mentioned in the results section.

2.     In the Conclusion the authors refer to deforestation for the first time as a contributing factor for interaction between humans and animals. I suggest that the authors move this to the discussion because it is not a major finding of the results.

3.     Line 176: The tree should have at least a 1000 bootstrap repetitions

Minor corrections:

Line 32-33: There is general information about the samples but do the authors have specific information about date and place of the two samples? This should be included in the sample section.

Line 77: with one sequences

Line 199: one individuals… five individual

Line 201: These genome

Line 203: de novo assemble

Line 205: related to

Line 211: please also give the length of the segment which the similarity is based on

Line 261-263: Please explain or rewrite

Table 1: Please correct the formatting of the table

Line 282: divergent divergence

Line 292-293: to nucleotide identity

Lines 303-306: Please rewrite to two sentences

Author Response

The authors identify and describe the genetic make up of two CRESS strains detected in in children with gastrentiritis. These strains are related to Howler monkey associated porprismacovirus. The manuscript is well written, and the data supports the hypothesis. My main comment is on the consistency between the different sections of the paper.

 The authors should pay special attention to two things:

1.     In the discussion (lines 310 – 313) the authors comment on two trees based on the proteins of CAP and REP. However, in the results section the only tree discussed (Fig 2) is based on the whole genome. I didn’t see any supplementary data and if these trees were done it should be mentioned in the results section.

Resp: In this version of the manuscript, we have included in the supplementary material a figure that shows CAP and REP trees.

2.     In the Conclusion the authors refer to deforestation for the first time as a contributing factor for interaction between humans and animals. I suggest that the authors move this to the discussion because it is not a major finding of the results.

Resp: In the new version of the manuscript, we changed this.

3.     Line 176: The tree should have at least a 1000 bootstrap repetitions

 Resp:. The boostrap test in the programme used to build all trees has this as its default value. When sequences are highly divergent, we utilized the Shimodaira-Hasegawa test, which outperforms bootstrap.

Minor corrections:

Line 32-33: There is general information about the samples but do the authors have specific information about date and place of the two samples? This should be included in the sample section.

 Resp: In a previous descriptive paper (Viruses 2021, 13, 2365, doi:10.3390/v13122365), we characterised all samples in detail and provided their locations. We have summarized the material from the samples in the current publication so that it is not repetitive. All pertinent information, including location, age, gender, and clinical status, can be found in reference 34's supplementary material.

Line 77: with one sequences

Resp: This was changed in the manuscript

Line 199: one individuals… five individual

Resp: This is was also changed in the new manuscript

Line 201: These genome

Resp: Changed in the manuscript

Line 203: de novo assemble

Resp: The sentence was modified in the revised manuscript.

Line 205: related to

Resp: This was changed to related with...

Line 211: please also give the length of the segment which the similarity is based on

Resp: This information was included in Table 1.

Line 261-263: Please explain or rewrite

Resp: We changed this sentence.

Table 1: Please correct the formatting of the table

Resp: This table's format has been changed.

Line 282: divergent divergence

Resp: This was changed in the revised manuscript

Line 292-293: to nucleotide identity

 Resp: This is genetic divergence and not nucleotide identity.

Lines 303-306: Please rewrite to two sentences

Resp: Not sure was wrong with the initial sentence in the discussion section.